# Antigen concentration, viral load, and test performance for SARS-CoV-2 in multiple specimen types

**Allison Golden** [1][☌]*, **Michelle Oliveira-Silva** [2][☌], **Hannah Slater** [1][☌], **Alexia Martines Vieira** [2], **Pooja Bansil** [1], **Emily Gerth-Guyette** [1], **Brandon T. Leader** [1], **Stephanie Zobrist** [1], **Alan Kennedy Braga Ferreira** [2], **Erika Crhistina Santos de Araujo** [2], **Catherine Duran de Lucena Cruz** [2], **Eduardo Garbin** [2], **Greg T. Bizilj** [1], **Sean J. Carlson** [1], **Mariana Sagalovsky** [1], **Sampa Pal** [1], **Vin Gupta** [3], **Leo Wolansky** [4], **David S. Boyle** [1], **Deusilene Souza Vieira Dall'Acqua** [5], **Felipe Gomes Naveca** [6], **Valdinete Alves do Nascimento** [6], **Juan Miguel Villalobos Salcedo** [5], **Paul K. Drain** [7,8], **Alexandre Dias Tavares Costa** [9], **Dhélio Pereira** [2], **Gonzalo J. Domingo** [1]

1 Diagnostics, PATH, Seattle, Washington, United States of America, 2 Centro de Pesquisa em Medicina Tropical (CEPEM), Porto Velho, Rondônia, Brazil, 3 Amazon.com, Seattle, Washington, United States of America, 4 Pandemic Prevention Institute, The Rockefeller Foundation, New York City, New York, United States of America, 5 Fundação Oswaldo Cruz (FIOCRUZ), Porto Velho, Rondônia, Brazil, 6 Instituto Leônidas e Maria Deane (ILMD), Fundação Oswaldo Cruz (FIOCRUZ), Manaus, Amazonas, Brazil, 7 Department of Global Health, University of Washington, Seattle, Washington, United States of America, 8 Department of Medicine, University of Washington, Seattle, Washington, United States of America, 9 Instituto Carlos Chagas (ICC), Fundação Oswaldo Cruz (FIOCRUZ), Curitiba, Paraná, Brazil

☌ These authors contributed equally to this work.
* algolden@path.org

**Data Availability Statement:** Study data will be made available online at Harvard Dataverse: https://doi.org/10.7910/DVN/NGNUXY.

## Abstract

The relationship between N-antigen concentration and viral load within and across different specimens guides the clinical performance of rapid diagnostic tests (RDT) in different uses. A prospective study was conducted in Porto Velho, Brazil, to investigate RDT performance in different specimen types as a function of the correlation between antigen concentration and viral load. The study included 214 close contacts with recent exposures to confirmed cases, aged 12 years and older and with various levels of vaccination. Antigen concentration was measured in nasopharyngeal swab (NPS), anterior nares swab (ANS), and saliva specimens. Reverse transcriptase (RT)–PCR was conducted on the NPS and saliva specimens, and two RDTs were conducted on ANS and one RDT on saliva. Antigen concentration correlated well with viral load when measured in the same specimen type but not across specimen types. Antigen levels were higher in symptomatic cases compared to asymptomatic/oligosymptomatic cases and lower in saliva compared to NPS and ANS samples. Discordant results between the RDTs conducted on ANS and the RT-PCR on NPS were resolved by antigen concentration values. The analytical limit-of-detection of RDTs can be used to predict the performance of the tests in populations for which the antigen concentration is known. The antigen dynamics across different sample types observed in SARS-CoV-2 disease progression support use of RDTs with nasal samples. Given lower antigen concentrations in saliva, rapid testing using saliva is

**Funding:** This work was supported by grants from The Rockefeller Foundation [2020 HTH 039] and Amazon.com [2D-04020007] to GJD. Rockefeller and Amazon's contributions to the study and publication are represented by authors VG and LW. FGN and VAN were supported by the National Council for Scientific and Technological Development [grant 403276/2020-9] and Inova Fiocruz / Fundação Oswaldo Cruz [grant VPPCB-007-FIO-18-2-30 - Knowledge generation]. FGN is a National Council for Scientific and Technological Development (CNPq) fellow. Benchmarking work cited and that was used in analysis was funded by the Bill & Melinda Gates Foundation (https://www.gatesfoundation.org/) via grant INV-016821. Other than the contributions by authors VG and LW, funders did not have any role in the study design, data collection and analysis, decision to publish, or preparation of the manuscript.

**Competing interests:** The authors have declared that no competing interests exist.

expected to require improved RDT analytical sensitivity to achieve clinical sensitivity similar to rapid testing of nasal samples.

## Introduction

Diagnostic testing has a critical role in the public health response to the severe acute respiratory syndrome coronavirus 2 (SARS-CoV-2) pandemic [1]. Early detection of the virus can help to limit transmission, inform infection-control measures, and guide appropriate clinical management of patients. The reference standard for SARS-CoV-2 testing is molecular detection of viral RNA with reverse transcriptase (RT)–PCR assays [2]. However, this laboratory-based method is technically complex, costly, and requires robust specimen transport and reporting systems–all of which limit its utility and availability in many settings [3]. Since the onset of the pandemic, an unprecedented number of rapid diagnostic tests (RDTs) have been developed to address this gap, including many designed to detect viral antigens [1]. Such tests play important roles in increasing access to SARS-CoV-2 testing globally.

Viral load varies across different specimen types throughout the progression of COVID-19 disease and between individuals [4, 5]. The variant and vaccination status may also influence the relative viral loads [6, 7]. However, the performance of RDTs has often been compared to RT-PCR results conducted on nasopharyngeal swab (NPS) specimens as a gold standard, and was initially the regulatory reference, regardless of the type of specimen that the RDT is conducted on. The majority of RDT used to screen for COVID-19 infection do so by detecting the nucleocapsid or N-antigen. Understanding RDT performance relative to RT-PCR may be limited by the relationship between the antigen concentration in the sampled oral or nasal cavity and viral copies in the NPS. Antigen concentration dynamics in different specimen types can inform how RDTs can be expected to perform in different stages of infection and use-case scenarios.

In 2021, a prospective diagnostic accuracy study was conducted among close contacts of COVID-19-positive index cases in Porto Velho, Brazil [8]. This study evaluated the performance of four tests for SARS-CoV-2—including three rapid antigen tests (STANDARD Q COVID-19 Ag Nasal and Saliva tests [SD Biosensor, Republic of Korea] and the SARS-CoV-2 Ag Test [LumiraDx™ Limited, United Kingdom]), as well as one dual-plexed RT-PCR method that uses minimally processed saliva [9, 10] (SalivaDirect™ protocol [Yale University, United States])—against a regulatory-approved RT-PCR assay run on NPS specimens. Here, we present the results of reference antigen concentration measurements on various specimen types collected from this study and determine the correlations between antigen concentration and viral load.

## Materials and methods

### Study design and clinical procedures

The design of this clinical study has been described previously [8]. Briefly, between July and September 2021, a prospective diagnostic accuracy study was conducted among close contacts of 50 confirmed COVID-19-positive index cases in Porto Velho, Brazil. Close contacts (aged 12 years and older) were identified through contact elicitation interviews administered to index cases. All eligible close contacts were invited to participate in the study. A subset of those who shared a primary residence with the index case were followed longitudinally for clinical evaluations and testing every other day for up to five visits total (up to eight additional days after the initial visit), or until the participant obtained a positive result on the rapid test

administered during the visit. In such cases, only one additional visit was performed, but an NPS was not collected to minimize staff exposure.

Information on medical history, participant demographics, and symptoms were collected at enrollment, with the latter also collected at each study visit. Additionally, at each visit, two paired ANS, one NPS, and one saliva sample, from passive saliva that pools naturally in the mouth, were collected. The details of specimen collection have been previously published (supplementary materials in Zobrist et al., 2022 [8]). The STANDARD Q COVID-19 Ag Nasal Test was run on one ANS during the visit. The remaining lysis buffer from this test was retained for antigen testing and remaining samples were transferred to the laboratory to be stored frozen.

### Tests evaluated

Three antigen tests were evaluated: the STANDARD Q COVID-19 Ag Nasal Test and LumiraDx SARS-CoV-2 Ag Test on ANS samples and the STANDARD Q COVID-19 Ag Saliva Test on saliva samples. One molecular test was also evaluated: the SalivaDirect protocol, a dualplexed RT-PCR method for SARS-CoV-2 detection from minimally processed saliva [10].

### Specimen characterization summary

RT-PCR and SalivaDirect RT-PCR were conducted on the NPS and saliva specimens, respectively. Quantitative viral load values are available for the NPS sample, while only Ct values are available for the saliva sample. Rapid antigen tests were conducted on ANS and saliva samples. The antigen concentration was determined in all three specimen types: for NPS it was measured from VTM, for ANS from the same STANDARD Q COVID-19 Ag lysis buffer used for the corresponding antigen test, and for saliva directly from saliva.

### Laboratory procedures

The saliva samples and the second ANS, extracted with LumiraDx buffer, were stored on ice following collection and were frozen immediately on return to the laboratory within five hours of collection. The LumiraDx SARS-CoV-2 Ag Test and the STANDARD Q COVID-19 Ag Saliva Test were both run using thawed specimens within five days after initial freezing of the samples. The SalivaDirect assay was run in batches. All tests were run by operators blinded to point-of-care and reference results.

### Reference testing

Reference testing was conducted on NPS with a multiplex real-time PCR assay (Allplex™ SARS-CoV-2 Assay [Seegene Inc., Republic of Korea]). A CFX96 real-time PCR machine (Bio-Rad, United States) [11] was used, and an automated RNA extraction was conducted using the Extracta kit (Loccus, Brazil). All SARS-CoV-2-positive specimens were repeated on the same assay for quantitative estimation of viral load using a dilution series of Amplirun® Coronavirus RNA Control (Vircell S.L., Spain) for viral load quantification on each plate. The limit of quantification was 125 copies/mL.

### Genomic sequencing

Specimens with cycle threshold (Ct) values <30 underwent genomic sequencing [8]. Genomic sequencing for this study was conducted using Illumina COVIDSeq® Test (Illumina Inc., United States), with some modifications [12], on Illumina's MiSeq® or NextSeq 1000® sequencers. Reads were assembled using DRAGEN COVID Lineage App, v. 3.5.4, at Illumina

BaseSpace (http://basespace.illumina.com) or BBMap 38.84 embedded in Geneious Prime 2022.1 (https://www.geneious.com). Consensus sequences were analyzed for quality issues with Nextclade (https://clades.nextstrain.org) [13], and lineages were identified using the pangolin tool on its most up-to-date version [14]. All SARS-CoV-2 genomes generated and analyzed in this study are available at the EpiCoV database in GISAID (https://www.gisaid.org) [15].

## Measuring antigen concentration

SARS-CoV-2 N concentration was measured using the Meso Scale Discovery platform (Meso Scale Diagnostics, United States), which uses electrochemiluminescence for detection. The assay details have been described previously [16]. Specimens were shipped to PATH (Seattle, WA, USA) where the antigen concentration assay was conducted. The SARS-CoV-2 nucleo-capsid protein quantitative immunoassay on the platform was run using a 10-point recombinant protein standard curve ranging from 0.128 pg/mL to 100 ng/mL for quantification. Briefly, 25 uL of neat or diluted viral transport media (VTM) from the eluted NPS, STANDARD Q COVID-19 Ag lysis buffer ANS, or saliva sample were diluted in an equal volume of 2x running buffer for samples for a final sample dilution of 2-fold and final detergent concentration of 1%. Neat samples were used for those testing negative on cognate assays, or with Ct values on the NPS of greater than 25. For NPS specimens with Ct values between 21 and 25 and presumed positives without Ct information, a sample five-fold dilution was performed, and for those with Ct values of 20 or less, a sample twenty-fold dilution was performed to ensure the results were within the dynamic range of the assay. Sample dilution of presumed antigen-positive ANS were five or twenty-fold depending on combination of assay results and NPS results. Saliva specimens were diluted ten-fold if both SalivaDirect and the STANDARD Q COVID-19 Ag Saliva Test were positive. Positive specimens were run in duplicate and discrepant results of coefficient of variation over 20%, or those outside the assay detection range, were repeated. Samples exceeding the dynamic range of the assay were repeated with increased dilution. Samples which were diluted based on expected concentration and had no antigen detected were rerun at the lowest dilution of two-fold. Samples which tested positive for antigen but were expected to be negative were repeated to confirm. Duplicate well quality control samples consisting of previously aliquoted, diluted recombinant protein at 3 different concentrations were run with each plate. Across 21 plates run, all controls were within the median +/-20% for concentration and no controls had CVs over 20%.

## Statistical analysis

Sensitivity, specificity, and positive and negative predictive values were calculated using standard formulas and presented with 95% confidence intervals. Correlations between antigen values, viral load (viral genome equivalents) values and SalivaDirect Ct values were assessed using the R-squared value generated from a linear model fitted to log10 transformed data (apart from the SalivaDirect Ct data which were not transformed). Differences between distributions of antigen concentrations based on symptom status were calculated using a two-sided non-paired t-test. The relationships between antigen concentration or viral load and the probability of positivity by each test were modeled using a logistic regression model fitted using the R package brms [17]. Gaussian priors were used for the parameters and models were run for 10,000 iterations to ensure convergence. Data were collected and managed using REDCap electronic data capture tools [18]. Statistical analyses were conducted using R 4.0.3 (R Foundation for Statistical Computing, Austria). Study data will be made available online at Harvard Dataverse: https://doi.org/10.7910/DVN/NGNUXY

### Ethical considerations

WCG Institutional Review Board (1301165), the CEPEM ethics committee, and Brazil's National Research Ethics Commission approved this study (44351421.0.0000.0011). Written informed consent was obtained for all participants.

## Results

### Participant characteristics

In total, 214 close contacts were enrolled from 50 symptomatic COVID-19-confirmed index cases. Of these 214 close contacts, 64 shared a primary household with their associated index case and were followed longitudinally, and 150 were non-household contacts which included friends, family in other houses, classmates, coworkers, neighbors, and other contacts of the index case. Of the 64 household contacts, 35 tested positive by reference NPS PCR during at least one of their visits. Of the 150 non-household contacts, 29 tested positive by reference NPS PCR during their visit. Study participants were classified as either symptomatic, oligo-symptomatic, or asymptomatic, as described in S1 Table. Full details of the study participant demographics, including vaccination status, have been previously published [8].

### Measurement of antigen concentration in clinical samples

Antigen concentration was measured in all three available specimen types for each participant and timepoint combination selected from the total based on the following: those participants with at least one positive test result on any of the molecular or antigen detection tests (n = 166) at that time point, participant time points which were negative for all results at a time point only but for participants which at some point had a positive test result (n = 33), and participant time points which were from participants who had never had any positive result on any test for any time point (n = 136). In total, antigen concentration was determined for 961 samples which were available from the selected participant time points and two additional saliva samples from negative participant time points (S1 Table).

### Relationship between antigen concentration and symptoms

Antigen concentration in specimens from symptomatic individuals (n $\geq$ 84) was significantly higher than in specimens from asymptomatic or oligosymptomatic study participants (n $\geq$ 39) across all clinical specimen types (P<0.01) (Fig 1). Antigen concentration was also significantly lower in saliva specimens as compared to ANS and NPS specimens (P<0.01), with the mean values for symptomatic saliva specimens being almost two orders of magnitude lower than those for the ANS and NPS specimens.

### Relationship between viral load and antigen concentration

A good correlation between antigen concentration and viral load (or Ct value on the SalivaDirect assay) was observed between measurements conducted on cognate specimens: [virus]$_{NPS}$/[N antigen]$_{NPS}$ ($R^2$ = 0.721) and [Ct]$_{saliva}$/[N-antigen]$_{saliva}$ ($R^2$ = 0.725) (Fig 2). Antigen concentration measurements in NPS were positive for detectable antigen in 77.5% of the paired RT-PCR positives, with undetectable antigen only in lower viral load samples. In contrast, poor correlation was observed for the same measurement across ANS specimens relative to the NPS viral load: [virus]$_{NPS}$/[N-antigen]$_{ANS}$ ($R^2$ = 0.471). Very poor correlation was observed across NPS and saliva samples: [virus]$_{NPS}$/[N-antigen]$_{saliva}$ ($R^2$ = 0.061).

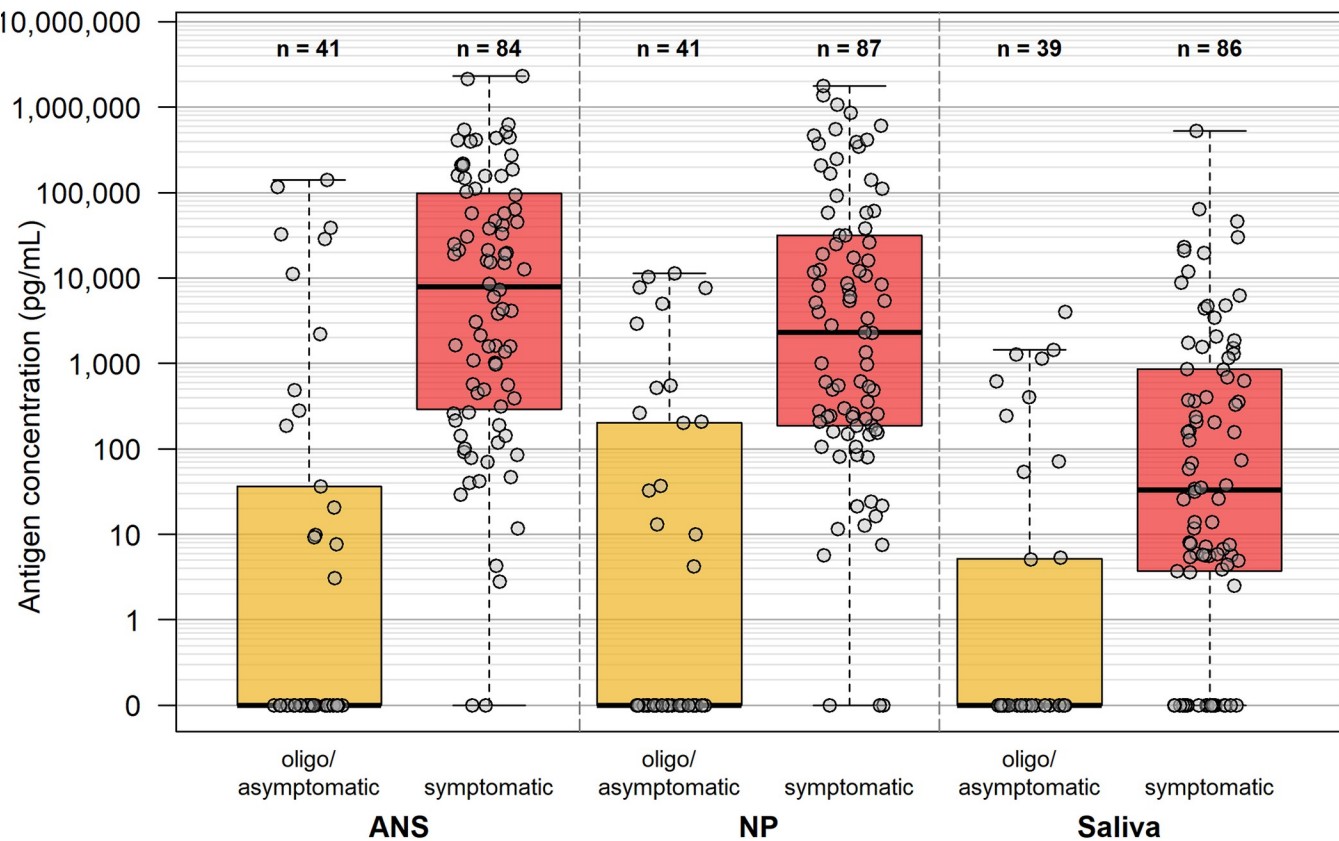

**Fig 1. Box and whisker plots showing antigen concentration distributions among symptomatic cases and oligo/asymptomatic cases among the close contacts in anterior nares swab (ANS), nasopharyngeal swab (NPS), and saliva specimens.** The number (n) of specimens per category is listed above each category.

## Relationship between antigen concentration and antigen detection test result

An overall dose-response relationship is observed when relating antigen concentration to test signal intensity on both STANDARD Q COVID-19 Ag tests and on the LumiraDx SARS-CoV-2 Ag Test (Fig 3). For the STANDARD Q COVID-19 Ag tests, line signal intensity defined by a predefined scale (S1 Fig) rose with antigen concentration measured in the same specimen. Likewise, on the LumiraDx SARS-CoV-2 Ag Test, signal intensity as measured by the LumiraDx instrument (and provided by LumiraDx) rose with antigen concentration up to instrument signal saturation. The probability of test positivity plotted against antigen concentration in the ANS specimen for LumiraDx SARS-CoV-2 Ag Test and the STANDARD Q COVID-19 Ag tests conducted on ANS and against saliva antigen concentration for the STANDARD Q COVID-19 Ag Saliva Test show overlapping profiles indicative of similar analytical performance of the tests in their respective specimens (Fig 4A). The probability of positivity for the three tests against NPS viral load has broader 95% credible intervals and more gradual probability increases with increasing viral load concentrations, particularly for the STANDARD Q COVID-19 Ag Saliva Test (Fig 4B). A subset (N = 9) of participant had NPS specimens with Ct values below 28 yet had negative antigen rapid test results. Genome sequencing was available for 8 of these specimens. Sequencing did identify sense mutations on the N antigen on some of these discordant specimens, but the same mutations were also found

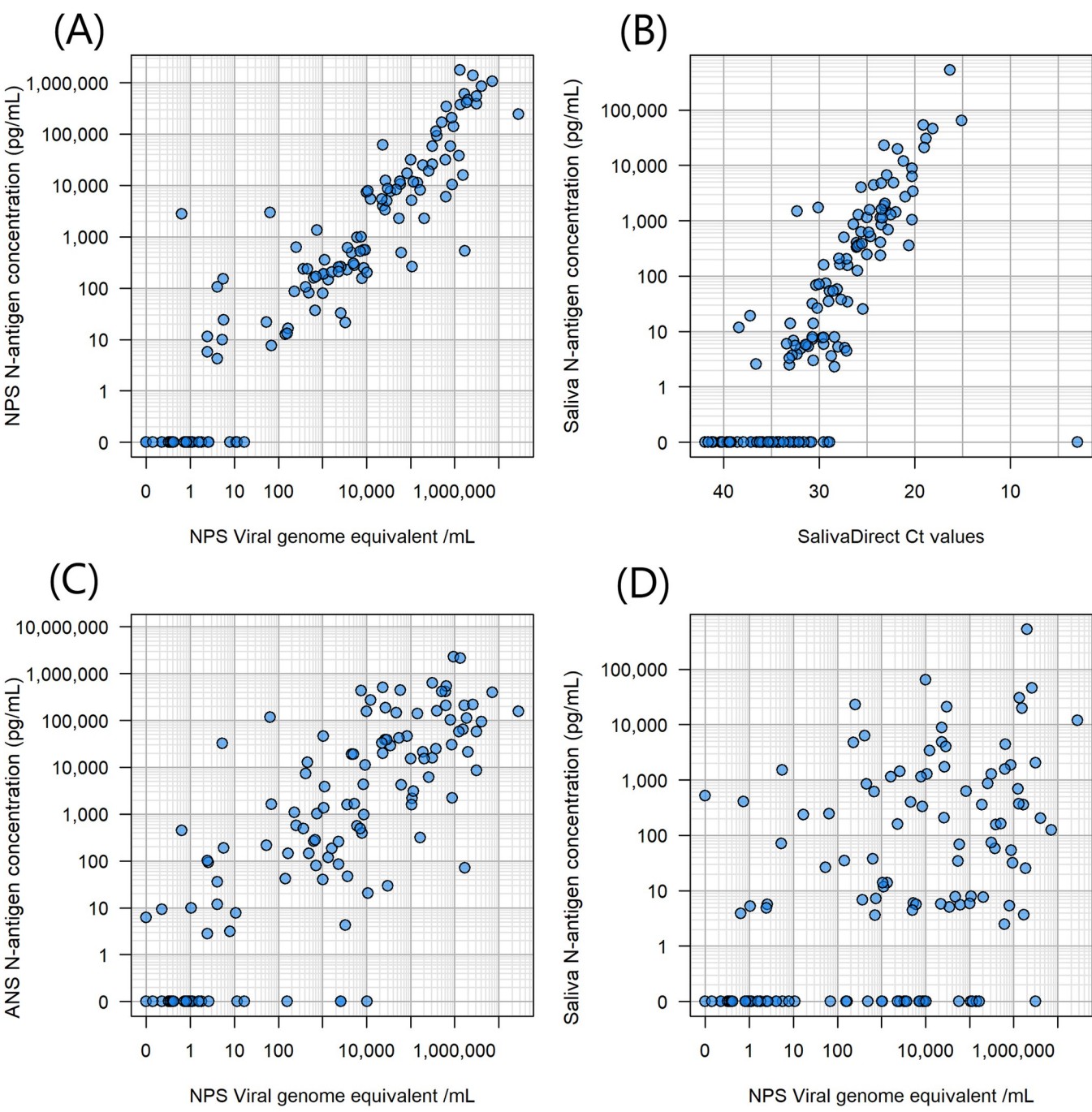

**Fig 2. Relationship between viral load and antigen concentration.** Viral load was measured from NPS stored in VTM. The Saliva Direct assay provides Ct values in the saliva specimen. Antigen concentration was measured from the same NPS-VTM specimen, from ANS specimen stored in the STANDARD Q lysis buffer and from saliva. Correlations are shown between (A) NPS viral load and NPS antigen concentration, (B) saliva viral load (in Ct values) and saliva antigen concentration, (C) NPS viral load and ANS antigen concentration, and (D) NPS viral load and saliva concentration.

on specimens where there were no discordant results (S3 Table). Including also participant samples for which the NPS PCR result was Ct below 34 yet had one or more negative rapid test result (n = 14), antigen concentrations in the ANS specimen extracted in STANDARD Q COVID-19 Ag lysis buffer for these participants were found to be low or undetectable.

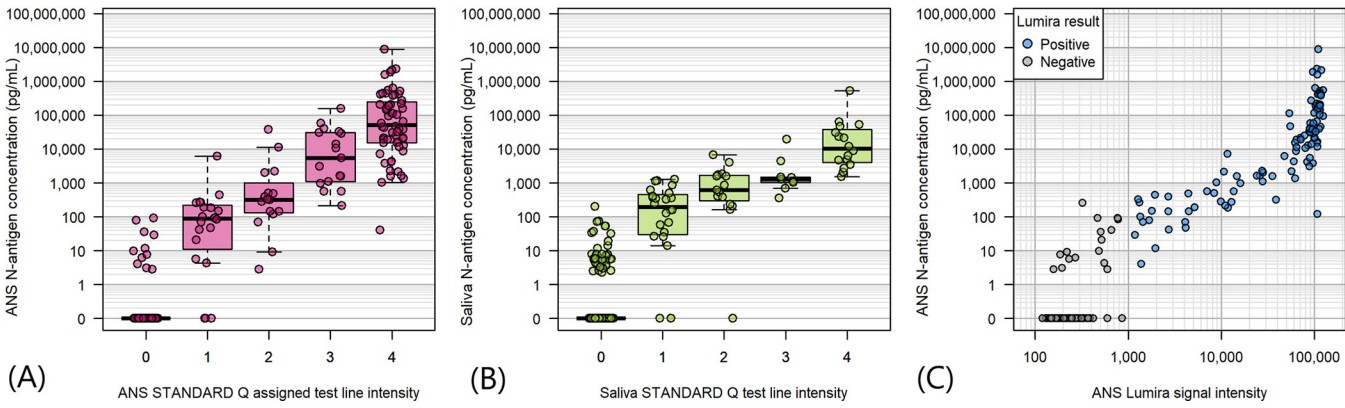

**Fig 3. Relationship between N-antigen signal and rapid antigen diagnostic signal.** Antigen concentration was measured from an anterior nasal swab specimen stored in the STANDARD Q lysis buffer and from saliva. The relationships are shown for Panel (A) the ANS STANDARD Q test signal and the ANS N-antigen concentration, Panel (B) the saliva STANDARD Q assigned test signal and the saliva N-antigen concentration, and Panel (C) the ANS Lumira signal and ANS N-antigen concentration.

Without clear mutation association and with overall low antigen concentrations found in the discordant ANS the results suggest that false negatives to PCR by rapid test were most likely due to low antigen availability in the ANS specimens.

**Validation of predictive performance modeling from analytical sensitivity results.**
Both the STANDARD Q COVID-19 Ag and LumiraDx SARS-CoV-2 Ag tests had previously been evaluated for their analytical performance against antigen concentration using a clinical specimen pool in a laboratory [19]. The antigen limits of detection, as defined by antigen concentrations at which tests were expected to be positive 90% of the time, were applied as

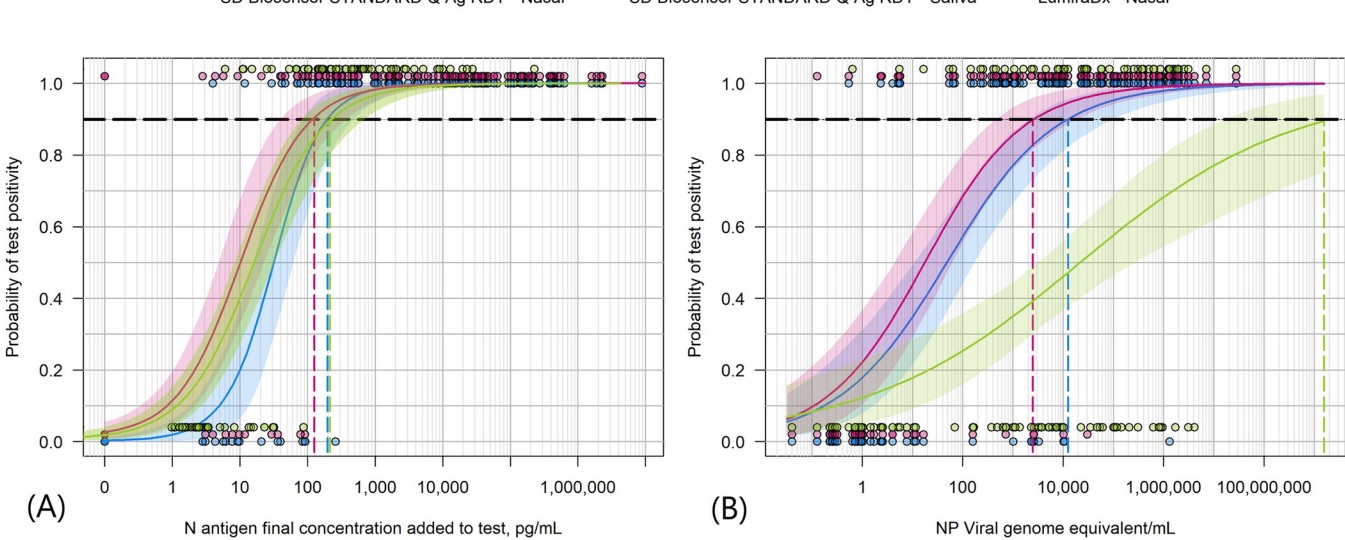

**Fig 4. Probability of positivity for the rapid antigen tests as a function of antigen concentration on their cognate specimen.** Points plot test results (1 = positive, 0 = negative) versus concentration of N antigen in sample. Lines represent fits to data points to model probability of positivity. Panel (A): Probability for positive test result for Lumira (blue) and the STANDARD Q test (magenta) conducted on the ANS specimen as a function of the antigen concentration in the ANS specimen, and the STANDARD Q test (green) conducted on the saliva specimen as a function of the antigen concentration in the saliva specimen. Panel (B): Probability of positive test for the same three tests as a function of viral load in NPS specimens. The antigen concentration or viral load at which there is greater than 90% probability of a positive test result is indicated. The shaded areas show the 95% credible intervals for the probability functions.

**Table 1. Sensitivity of the RDTs conducted on ANS samples from all close contacts.** The sensitivity was determined against confirmed positive cases by RT-PCR on NPS. The table presents (i) the observed sensitivity of the antigen detection tests performed on ANS and (ii) their predicted performance based on analytical limit-of-detection.

| | ANS sample (all) | | ANS sample (Ct below 34) | |
|---|---|---|---|---|
| | STANDARD Q COVID-19 Ag | SARS-CoV-2 Ag Test [LumiraDx] | STANDARD Q COVID-19 Ag | SARS-CoV-2 Ag Test [LumiraDx] |
| Observed sensitivity of antigen tests in close contacts | 55.0 (44.1–66.9) N = 79 | 50.6 (39.1–62.1) N = 78 | 84.0 (70.9–92.8) N = 49 | 79.6 (65.7–89.8) N = 50 |
| Predicted sensitivity of antigen tests in close contacts based on antigen concentration and analytical limits of detection [19] | 50.0 (38.5–61.5) N = 78 | 50.0 (38.5–61.5) N = 78 | 79.6 (65.7–89.8) N = 49 | 79.6 (65.7–89.8) N = 49 |

thresholds in the measured antigen concentration in the ANS specimens and used to categorize the predicted test results as detected or not by each rapid test. Clinical performance of the two tests among the close contacts in the study was then predicted based on comparison of these results to the gold standard of NPS RT-PCR positive cases.

Table 1 shows the actual observed sensitivity and the predicted sensitivity in this study population composed of symptomatic and asymptomatic/oligosymptomatic cases based on analytical performance and antigen concentrations.

## Discussion

This study investigated the relationship between antigen concentration and diagnostic test performance on different types of samples collected in a study conducted in Porto Velho, Brazil, in 2021 [8]. The study shows that antigen concentration is closely related to viral load when measured from the same specimen type. The NPS antigen concentration corresponded to NPS viral load, and the saliva antigen concentration measured in saliva samples correlated with Ct values from the SalivaDirect assay, which has been previously observed [20, 21]. However, the antigen concentrations in ANS and saliva specimens did not correlate well with the viral load measured in the same study participants' NPS specimens. N-antigen concentrations are reflective of virus levels in the corresponding oral/nasal cavities from which they are collected but do not reflect viral levels in other oral/nasal cavities.

The antigen concentration corresponded very well with both the signal intensity and test positivity of the rapid antigen detection tests within the same sample type. As demonstrated by overlapping curves, no significant difference in the probability of test positivity as a function of cognate sample type antigen concentration was observed across the three tests included in this study: the STANDARD Q COVID-19 Ag conducted on ANS and saliva and the LumiraDx SARS-CoV-2 Ag Test conducted on ANS. In contrast, when test positivity was modeled as a function of viral copy number in the NPS, the performance of the saliva antigen test was significantly lower; this is expected based on the poor correlation between saliva antigen concentration and NPS viral load. Additionally, "discordant" specimens (with high NPS viral loads defined by Ct values of <34, but negative antigen test results on ANS) were resolved by confirming that, in the ANS specimen, there was indeed low or undetectable N-antigen levels. The false negative results could not be attributed to variations in the N gene and test failure, as the N gene mutations detected in the study were shared by other samples with concordant results.

These observations are important when considering study designs for antigen detection test evaluations and assessing their performance. From a regulatory perspective, NPS RT-PCR has been considered the gold standard for SARS-CoV-2 diagnostic test evaluation, regardless of the specimen source for the index test. This study shows that discordant results can and should be anticipated if the specimen source for the RDT under evaluation is different to the specimen used to measure viral load. Within a specimen type, antigen concentration, and therefore

antigen detection test performance, can be expected to relate to viral load. These results may also explain some of the differences in performance observed for SARS-CoV-2 tests across different sample types [22–24].

The study confirmed higher antigen concentrations in symptomatic cases compared to asymptomatic and oligosymptomatic cases across all three specimen types, as anticipated given the association between viral carriage and symptom onset. While recent studies suggest that early onset of SARS-CoV-2 can be first observed in the saliva by RT-PCR [4, 5, 21], the overall lower antigen levels observed in saliva in this study suggests that current rapid antigen tests are not sensitive enough to leverage this opportunity for earlier infection detection.

Combining the laboratory analytical limits-of-detection determined from a clinical specimen pooled sample with the distribution of antigen concentrations in this study population, it was possible to estimate the observed performance of these antigen detection tests in this study population This result confirms that benchmarking analytical performance can be related to clinical performance, albeit with caution. Larger data sets of paired viral load and antigen concentration measurements across different sample types, across variants, and in different study populations may improve the ability to predict the performance of antigen detection tests in different scenarios and use cases. Of note, the population included both symptomatic cases and asymptomatic/oligosymptomatic cases, including samples with low viral load counts and undetectable antigen concentration.

Some limitations to the findings of this study are that (a) RT-PCR was not conducted on the ANS samples, (b) RDTs were not conducted on NPS samples, (c) the SalivaDirect test was not conducted as a quantitative test, and (d) antigen concentration was measured across different matrices from samples collected using different swabs, potentially leading to inherent biases or systematic errors in the antigen quantification. In particular, the ANS antigen concentration was determined from the same extracted specimen from which the STANDARD Q COVID-19 Ag Test was conducted. In contrast, the LumiraDx SARS-CoV-2 Ag Test was conducted on a second ANS sample taken in parallel that had been extracted in its corresponding buffer and subsequently frozen according to instructions for use, prior to testing. Dilutional differences from extraction buffer volumes were accounted for in the predicted sensitivity, but matrix, swab, and sampling handling differences remain. While strong antigen to viral load relationships were observed within NPS and saliva samples, poorer correlation was observed between ANS antigen and NPS viral load. This is a limitation in the interpretation of performance of the tests using ANS in terms of sensitivity to viral load.

In conclusion, this study indicates the value of understanding the underlying antigen concentration dynamics and its relationship to viral load across different sample types to inform and predict RDT performance in different settings, variants, and use cases.

## Supporting information

**S1 Table. Case definitions for the study.** Descriptions are of participant symptoms, regardless of test positivity.
(DOCX)

**S2 Table. Summary of SARS-CoV-2 cases, associated specimens, and number of antigen results.** Result totals are shown for the study participants for index cases, household contacts, and non-household contacts. The proportion of samples selected and run for antigen testing from each participant is shown for nasopharyngeal swab (NPS), anterior nares swab (ANS), and saliva. The lysis buffer for the STANDARD Q ANS specimen was used for antigen concentration determination in the ANS specimen. Because the 64 household contacts had multiple timepoints per individual, test results are reported for the combined individual and timepoint

together for a total of 224 samplings). Positive and negative classifications correspond to available test results from laboratory PCR results, SalivaDirect, STANDARD Q Saliva, LumiraDx, and STANDARD Q point-of-care (anterior nasal) and exclude antigen concentration measurement results. Totals are listed with breakdowns from those totals of number of Sympomatic (S), Oligosymptomatic (O), Asymptomatic (A).
(DOCX)

**S3 Table. Discrepancy analysis of specimens with a high NPS viral load but a negative test result from the antigen test conducted on the ANS specimen.** For binary test results, 1 = positive, 0 = negative. Test line intensity for STANDARD Q Nasal and Saliva tests were reported according to 0–4 scale shown in S1 Fig. Lumira Signal refers to signal output from the LumiraDx instrument.
(DOCX)

**S1 Fig. Score card for line intensity on the STANDARD Q COVID-19 Ag test (0 refers to no visible test line, or negative score) for nasal and saliva.**
(DOCX)

## Acknowledgments

The authors would like to thank all study participants, as well as the clinical and laboratory staff at CEPEM involved with this study. We also thank the team at Global Health Strategies for their contributions to this study, as well as SD Biosensor and LumiraDx for facilitating the availability of their tests for this study. Finally, we thank Amanda Tsang and Mara Lavery for editorial support with the manuscript.

## Author Contributions

**Conceptualization:** Emily Gerth-Guyette, Brandon T. Leader, Vin Gupta, Leo Wolansky, David S. Boyle, Paul K. Drain, Dhélio Pereira, Gonzalo J. Domingo.

**Data curation:** Michelle Oliveira-Silva, Hannah Slater, Alexia Martines Vieira, Pooja Bansil, Stephanie Zobrist, Greg T. Bizilj, Sean J. Carlson, Sampa Pal, Felipe Gomes Naveca, Valdinete Alves do Nascimento, Alexandre Dias Tavares Costa.

**Formal analysis:** Allison Golden, Michelle Oliveira-Silva, Hannah Slater, Pooja Bansil, Stephanie Zobrist.

**Funding acquisition:** Mariana Sagalovsky, Vin Gupta, Leo Wolansky, David S. Boyle, Paul K. Drain, Dhélio Pereira, Gonzalo J. Domingo.

**Investigation:** Allison Golden, Michelle Oliveira-Silva, Alexia Martines Vieira, Alan Kennedy Braga Ferreira, Erika Crhistina Santos de Araujo, Greg T. Bizilj, Sean J. Carlson, Felipe Gomes Naveca, Valdinete Alves do Nascimento, Alexandre Dias Tavares Costa.

**Methodology:** Allison Golden, Michelle Oliveira-Silva, Hannah Slater, Pooja Bansil, Emily Gerth-Guyette, Brandon T. Leader, Stephanie Zobrist, Sampa Pal, Deusilene Souza Vieira Dall'Acqua, Felipe Gomes Naveca, Juan Miguel Villalobos Salcedo, Paul K. Drain, Alexandre Dias Tavares Costa, Dhélio Pereira, Gonzalo J. Domingo.

**Project administration:** Michelle Oliveira-Silva, Stephanie Zobrist, Catherine Duran de Lucena Cruz, Eduardo Garbin, Mariana Sagalovsky, David S. Boyle, Deusilene Souza Vieira Dall'Acqua, Juan Miguel Villalobos Salcedo, Alexandre Dias Tavares Costa, Dhélio Pereira, Gonzalo J. Domingo.

**Software:** Hannah Slater.

**Supervision:** Allison Golden, Michelle Oliveira-Silva, Alexia Martines Vieira, Emily Gerth-Guyette, Sampa Pal, David S. Boyle, Deusilene Souza Vieira Dall'Acqua, Felipe Gomes Naveca, Juan Miguel Villalobos Salcedo, Alexandre Dias Tavares Costa, Dhélio Pereira, Gonzalo J. Domingo.

**Validation:** Allison Golden, Michelle Oliveira-Silva, Hannah Slater, Pooja Bansil, Stephanie Zobrist, Sampa Pal, Alexandre Dias Tavares Costa.

**Visualization:** Allison Golden, Michelle Oliveira-Silva, Hannah Slater, Gonzalo J. Domingo.

**Writing – original draft:** Allison Golden, Hannah Slater, Gonzalo J. Domingo.

**Writing – review & editing:** Allison Golden, Michelle Oliveira-Silva, Hannah Slater, Emily Gerth-Guyette, Stephanie Zobrist, Vin Gupta, Leo Wolansky, David S. Boyle, Paul K. Drain, Alexandre Dias Tavares Costa, Dhélio Pereira, Gonzalo J. Domingo.

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
