## [Decision Letter · Decision Letter 0]

20 Feb 2023

PONE-D-23-01146Antigen concentration, viral load, and test performance for SARS-CoV-2 in multiple specimen typesPLOS ONE

Dear Dr. Golden,

Thank you for submitting your manuscript to PLOS ONE. After careful consideration, we feel that it has merit but does not fully meet PLOS ONE’s publication criteria as it currently stands. Therefore, we invite you to submit a revised version of the manuscript that addresses the points raised during the review process.

We look forward to receiving your revised manuscript.

Kind regards,

Md Maruf Ahmed Molla

Academic Editor

PLOS ONE

Journal Requirements:

"This work was supported by grants from The Rockefeller Foundation [2020 HTH 039] and Amazon.com [2D-04020007] to GJD.  Rockefeller and Amazon's contributions to the study and publication are  represented by authors VG and LW.  FGN and VAN were supported by the National Council for Scientific and Technological Development [grant 403276/2020-9] and Inova Fiocruz / Fundação Oswaldo Cruz [grant VPPCB-007-FIO-18-2-30 - Knowledge generation].  FGN is a National Council for Scientific and Technological Development (CNPq) fellow. Benchmarking work cited and that was used in analysis was funded by the Bill & Melinda Gates Foundation (https://www.gatesfoundation.org/) via grant INV-016821. Other than the contributions by authors VG and LW, funders did not have any role in the study design, data collection and analysis, decision to publish, or preparation of the manuscript."

We note that one or more of the authors is affiliated with the funding organization, indicating the funder may have had some role in the design, data collection, analysis or preparation of your manuscript for publication; Rockefeller Foundation, Amazon, Inova Fiocruz (Fundação Oswaldo Cruz)

In other words, the funder played an indirect role through the participation of the co-authors. If the funding organization did not play a role in the study design, data collection and analysis, decision to publish, or preparation of the manuscript and only provided financial support in the form of authors' salaries and/or research materials, please do the following:

(1) Review your statements relating to the author contributions, and ensure you have specifically and accurately indicated the role(s) that these authors had in your study. These amendments should be made in the online form.

(2) Confirm in your cover letter that you agree with the following statement, and we will change the online submission form on your behalf: 

"This work was supported by grants from The Rockefeller Foundation [2020 HTH 039] and Amazon.com [2D-04020007] to Gonzalo J Domingo. Felipe Gomes Naveca and Valdinete Alves do Nascimento were supported by the National Council for Scientific and Technological Development [grant 403276/2020-9] and Inova Fiocruz / Fundação Oswaldo Cruz [grant VPPCB-007-FIO-18-2-30 - Knowledge generation]. Felipe Gomes Naveca is a National Council for Scientific and Technological Development (CNPq) fellow. Benchmarking work cited and that was used in analysis was funded by the Bill & Melinda Gates Foundation(https://www.gatesfoundation.org/) via grant INV-016821. The Bill & Melinda Gates Foundation did not have any role in the study design, data collection and analysis, decision to publish, or preparation of the manuscript."

"This work was supported by grants from The Rockefeller Foundation [2020 HTH 039] and Amazon.com [2D-04020007] to GJD.  Rockefeller and Amazon's contributions to the study and publication are  represented by authors VG and LW.  FGN and VAN were supported by the National Council for Scientific and Technological Development [grant 403276/2020-9] and Inova Fiocruz / Fundação Oswaldo Cruz [grant VPPCB-007-FIO-18-2-30 - Knowledge generation].  FGN is a National Council for Scientific and Technological Development (CNPq) fellow. Benchmarking work cited and that was used in analysis was funded by the Bill & Melinda Gates Foundation (https://www.gatesfoundation.org/) via grant INV-016821. Other than the contributions by authors VG and LW, funders did not have any role in the study design, data collection and analysis, decision to publish, or preparation of the manuscript."

Reviewers' comments:

Reviewer's Responses to Questions

**Comments to the Author**

1. Is the manuscript technically sound, and do the data support the conclusions?

Reviewer #1: Yes

Reviewer #2: Partly

2. Has the statistical analysis been performed appropriately and rigorously? 

Reviewer #1: Yes

Reviewer #2: Yes

3. Have the authors made all data underlying the findings in their manuscript fully available?

Reviewer #1: Yes

Reviewer #2: No

4. Is the manuscript presented in an intelligible fashion and written in standard English?

Reviewer #1: No

Reviewer #2: Yes

5. Review Comments to the Author

Reviewer #1: Comments to the Author

In this manuscript Number PONE-D-23-01146 entitled"Antigen concentration, viral load, and test performance for SARS-CoV-2 in multiple specimen types", the authors provided an overview of the SARS-CoV-2 diagnostic techniques within different specimens in Brazil with their clinical manifestations. This paper presents an interesting topic with a broad appeal to readers and presents a balanced view. However, it can be further improved and specific concerns need to be addressed before publication:

1. The title can be better modified to “The correlation between SARS-CoV-2 Antigen concentration and viral load using different techniques and specimens”.

2. 31-32 no need to repeat “specimen”.

3. 33 no need for “cases”.

4. 66 “The performance” correction.

5. 69,71 repeated word “understanding” please rephrase.

6. 86 “The design” correction.

7. 35, 88, 200, 201 can you please make it clear about the 214 cases and 50 cases in these positions, also in the supplementary table A? shouldn’t you have 214 cases and 65 positive cases, so what is this 50?

8. In table 1, can you address the symptomatic, asymptomatic/oligosymptomatic cases, as you mentioned in 274, 275

9. In Supplementary A, there are more than 65 positive cases of close contacts and in 202-203 “Among all the close contacts, 65 tested as SARS-CoV-2 positive by the reference assay at least once “, please make it clear.

Reviewer #2: The article titled 'Antigen concentration, viral load and test performance for SARS-CoV-2 in multiple specimen types' emphasizes on the comparison between the RT-PCR and RDT method. However, the importance of Nucleocapsid Protein o 'N Antigen' as the standard for viral load and antigen concentration measurement was not explained as there are other antigens such as 'ORF1b, S' that are also used in SARS-CoV-2 diagnostic methods. RT-PCR was not conducted on ANS sample and RDT was not conducted on NPS sample. As it was mentioned as a limitation, so how does this impact the findings of this study?

In page 8, the heading 'Antigen testing' is kind of misleading. it may be addressed as 'Rapid diagnostic testing/RDT of N antigen'. Also in page 10, the paragraph 'Antigen concentration determination' was not well versed. Table 1 needs to be rewritten and explained precisely. The grammatical and typographical errors needs to be corrected.

6. PLOS authors have the option to publish the peer review history of their article (what does this mean?). If published, this will include your full peer review and any attached files.

Reviewer #1: **Yes: **Ahmed H. Mousa

Reviewer #2: No

---

## [Author Response · Author response to Decision Letter 0]

24 May 2023

Dear Editors,

Thank you for the opportunity to revise the manuscript and we are grateful to the reviewers for their thoughtful review and hope we have addressed their comments.

Please find below the detailed replies in bold text below each comment.

Kind Regards,

Allison Golden

Journal Requirements:

Thank you, figure references and legend placement have been updated.

"This work was supported by grants from The Rockefeller Foundation [2020 HTH 039] and Amazon.com [2D-04020007] to GJD. Rockefeller and Amazon's contributions to the study and publication are represented by authors VG and LW. FGN and VAN were supported by the National Council for Scientific and Technological Development [grant 403276/2020-9] and Inova Fiocruz / Fundação Oswaldo Cruz [grant VPPCB-007-FIO-18-2-30 - Knowledge generation]. FGN is a National Council for Scientific and Technological Development (CNPq) fellow. Benchmarking work cited and that was used in analysis was funded by the Bill & Melinda Gates Foundation (https://www.gatesfoundation.org/) via grant INV-016821. Other than the contributions by authors VG and LW, funders did not have any role in the study design, data collection and analysis, decision to publish, or preparation of the manuscript."

We note that one or more of the authors is affiliated with the funding organization, indicating the funder may have had some role in the design, data collection, analysis or preparation of your manuscript for publication; Rockefeller Foundation, Amazon, Inova Fiocruz (Fundação Oswaldo Cruz)

In other words, the funder played an indirect role through the participation of the co-authors. If the funding organization did not play a role in the study design, data collection and analysis, decision to publish, or preparation of the manuscript and only provided financial support in the form of authors' salaries and/or research materials, please do the following:

(1) Review your statements relating to the author contributions, and ensure you have specifically and accurately indicated the role(s) that these authors had in your study. These amendments should be made in the online form.

(2) Confirm in your cover letter that you agree with the following statement, and we will change the online submission form on your behalf: 

Thank you for this, we have updated the Financial Disclosure to the following:

"This work was supported by grants from The Rockefeller Foundation [2020 HTH 039] and Amazon.com [2D-04020007] to GJD. Rockefeller and Amazon's contributions to the study and publication are represented by authors VG and LW. FGN and VAN were supported by the National Council for Scientific and Technological Development [grant 403276/2020-9] and Inova Fiocruz / Fundação Oswaldo Cruz [grant VPPCB-007-FIO-18-2-30 - Knowledge generation]. FGN is a National Council for Scientific and Technological Development (CNPq) fellow. Benchmarking work cited and that was used in analysis was funded by the Bill & Melinda Gates Foundation (https://www.gatesfoundation.org/) via grant INV-016821. The Bill & Melinda Gates Foundation did not have any role in the study design, data collection and analysis, decision to publish, or preparation of the manuscript. The authors VG and LW, contributed to the study design, decision to publish, and review of the manuscript."

"This work was supported by grants from The Rockefeller Foundation [2020 HTH 039] and Amazon.com [2D-04020007] to Gonzalo J Domingo. Felipe Gomes Naveca and Valdinete Alves do Nascimento were supported by the National Council for Scientific and Technological Development [grant 403276/2020-9] and Inova Fiocruz / Fundação Oswaldo Cruz [grant VPPCB-007-FIO-18-2-30 - Knowledge generation]. Felipe Gomes Naveca is a National Council for Scientific and Technological Development (CNPq) fellow. Benchmarking work cited and that was used in analysis was funded by the Bill & Melinda Gates Foundation(https://www.gatesfoundation.org/) via grant INV-016821. The Bill & Melinda Gates Foundation did not have any role in the study design, data collection and analysis, decision to publish, or preparation of the manuscript."

Please remove any funding-related text from the manuscript and let us know how you would like to update your Funding Statement. 

We have removed this language and the updated Financial Disclosure has been noted in the cover letter.

Please see above in our revised financial disclosure.

This has been done.

Reviewers' comments:

Reviewer #1: Comments to the Author

In this manuscript Number PONE-D-23-01146 entitled"Antigen concentration, viral load, and test performance for SARS-CoV-2 in multiple specimen types", the authors provided an overview of the SARS-CoV-2 diagnostic techniques within different specimens in Brazil with their clinical manifestations. This paper presents an interesting topic with a broad appeal to readers and presents a balanced view. However, it can be further improved and specific concerns need to be addressed before publication:

1. The title can be better modified to “The correlation between SARS-CoV-2 Antigen concentration and viral load using different techniques and specimens”.

Thank you for this suggestion, but after further internal consultation it was felt that the current title is most appropriate.

2. 31-32 no need to repeat “specimen”.

Thank you for this suggestion. We have updated the sentence to state the following:

The relationship between N-antigen concentration and viral load within and across different specimens guides the clinical performance of rapid diagnostic tests (RDT) in different uses.

3. 33 no need for “cases”.

Addressed above. 

4. 66 “The performance” correction.

The word “the” has been incorporated.

5. 69,71 repeated word “understanding” please rephrase.

Thank you. Removed the second “understanding”

6. 86 “The design” correction.

Changed “This” to “the”.

7. 35, 88, 200, 201 can you please make it clear about the 214 cases and 50 cases in these positions, also in the supplementary table A? shouldn’t you have 214 cases and 65 positive cases, so what is this 50?

This has been clarified in lines 201-207 and additionally in the Supporting information in Table S2.

8. In table 1, can you address the symptomatic, asymptomatic/oligosymptomatic cases, as you mentioned in 274, 275

Thank you for this suggestion. Because of the low numbers of cases, we have opted to present the performance across all close contacts with a confirmed RT-PCR result since separating these from the asymptomatic/oligosymptomatic cases results in broader confidence intervals. Breakdown of numbers of each category of patient disposition for each category are now listed in Table S2. Because asymptomatic cases overall showed lower levels antigen (Figure 1), it is expected performance would be lower in asymptomatic and oligosymptomatic cases given the correlations between viral load and N antigen concentrations in cognate specimens presented in this study.

9. In Supplementary A, there are more than 65 positive cases of close contacts and in 202-203 “Among all the close contacts, 65 tested as SARS-CoV-2 positive by the reference assay at least once “, please make it clear.

Thank you for pointing this out. We hope the new Table S2 and additional wording in lines 201-207 clarifies the distinction between all the numbers raised by the reviewer. Household close contact study participants may have multiple specimens associated. 

Reviewer #2: 

1.The article titled 'Antigen concentration, viral load and test performance for SARS-CoV-2 in multiple specimen types' emphasizes on the comparison between the RT-PCR and RDT method. However, the importance of Nucleocapsid Protein o 'N Antigen' as the standard for viral load and antigen concentration measurement was not explained as there are other antigens such as 'ORF1b, S' that are also used in SARS-CoV-2 diagnostic methods. 

Thank you for this. We have inserted the following statement in the introduction: ” The majority of RDT used to screen for COVID-19 infection do so by detecting the nucleocapsid or N-antigen.”

2.RT-PCR was not conducted on ANS sample and RDT was not conducted on NPS sample. As it was mentioned as a limitation, so how does this impact the findings of this study?

Thank you for this, 

• we have added the following limitation in line 335, “(b) RDTs were not conducted on NPS samples,”

• we added the following statement line 344: “While strong antigen to viral load relationships were observed in the NPS and saliva samples, this study did not directly measure the same relationship in ANS samples.”

3. In page 8, the heading 'Antigen testing' is kind of misleading. it may be addressed as 'Rapid diagnostic testing/RDT of N antigen'. 

Thank you for this. This section actually refers to determination of antigen concentration. So we have updated the section subheading to read: “Measuring antigen concentration”

4. Also in page 10, the paragraph 'Antigen concentration determination' was not well versed. 

We have changed this to measurement of “Antigen concentration in clinical samples”

5. Table 1 needs to be rewritten and explained precisely. 

Thank you, we have updated Table 1.

6. The grammatical and typographical errors needs to be corrected.

We have reviewed the article again and corrected any errors found.

 Have the authors made all data underlying the findings in their manuscript fully available?

Reviewer #1: Yes

Reviewer #2: No

Note to editors: All the study data minus the antigen concentration data is already available in dataverse as cited. On acceptance of the manuscript by the journal the spreadsheet will be updated to include the antigen concentration data.

---

## [Editor Report · Decision Letter 1]

13 Jun 2023

Antigen concentration, viral load, and test performance for SARS-CoV-2 in multiple specimen types

PONE-D-23-01146R1

Dear Dr. Golden,

We’re pleased to inform you that your manuscript has been judged scientifically suitable for publication and will be formally accepted for publication once it meets all outstanding technical requirements.

Kind regards,

Md Maruf Ahmed Molla

Academic Editor

PLOS ONE
---

## [Editor Report · Acceptance letter]

3 Jul 2023

PONE-D-23-01146R1 

Antigen concentration, viral load, and test performance for SARS-CoV-2 in multiple specimen types 

Dear Dr. Golden:

I'm pleased to inform you that your manuscript has been deemed suitable for publication in PLOS ONE. Congratulations! Your manuscript is now with our production department. 

Kind regards, 

on behalf of

Dr. Md Maruf Ahmed Molla 

Academic Editor

PLOS ONE